# Incidence of Rotational Malalignment after Intertrochanteric Fracture Intramedullary Nailing: A CT-Based Prospective Study

**DOI:** 10.3390/medicina60091535

**Published:** 2024-09-20

**Authors:** Michail Vavourakis, Dimitrios Zachariou, Athanasios Galanis, Panagiotis Karampinas, Meletis Rozis, Evangelos Sakellariou, Christos Vlachos, Iordanis Varsamos, John Vlamis, Elias Vasiliadis, Spiros Pneumaticos

**Affiliations:** 3rd Department of Orthopaedic Surgery, National & Kapodistrian University of Athens, KAT General Hospital, 14561 Athens, Greece; dimitriszaxariou@yahoo.com (D.Z.); athanasiosgalanis@yahoo.com (A.G.); karapana@yahoo.com (P.K.); mrozhs@gmail.com (M.R.); vagossak@hotmail.com (E.S.); christosorto@gmail.com (C.V.); jordan.var1995@gmail.com (I.V.); jvlamis@email.com (J.V.); eliasvasiliadis@yahoo.gr (E.V.); spirospneumaticos@gmail.com (S.P.)

**Keywords:** hip, hip anteversion, femur, femoral malrotation, rotational deformity, rotational overcorrection, prospective study, CT

## Abstract

*Background and Objectives:* Rotational malformation after intramedullary nailing of intertrochanteric fractures is a relatively common, possibly severe, and difficult-to-detect complication, since intraoperative radiographic imaging allows for the assessment of the quality of reduction in the frontal and sagittal planes, but not in the transverse plane. The purpose of this study is to evaluate the rotational malalignment after intramedullary nailing of intertrochanteric fractures and to investigate a possible connection with specific patients’ or fractures’ characteristics. *Materials and Methods:* 74 patients treated with intramedullary nailing due to an intertrochanteric fracture underwent a postoperative CT of the pelvis-hips and knees. The value of the anteversion was measured both in the operated-on (angle 1) and in the healthy hip (angle 2) based on the technique described by Jeanmart et al. and the difference in anteversion (D angle = angle 1 minus angle 2) was calculated. A positive D angle indicated the presence of excessive internal overcorrection of the distal fragment during fracture reduction, while a negative D angle indicated the presence of excessive external overcorrection. The absolute value of the D angle represents the postoperative difference in anteversion between the two hips. The patients were divided into three groups according to this value: group A, with D < 5° (physiological difference); group B, with 5° < D < 15° (acceptable rotational alignment); and group C, with D ≥ 15° (rotational deformity). *Results:* Group A constitutes 56.8%, group B 12.2%, and group C 31.1% of the study population. Overall, 79.7% of the patients presented a positive angle D, while, for group C, the percentage was even higher at 91.3%. According to the AO/OTA classification system, 37.8% of the cases were stable fractures, 47.3% were unstable fractures, and 14.9% were reverse oblique fractures. Based on our analysis, the type of fracture has a serious impact on the rotational alignment, since the statistical significance of the mean angle D for the three types of fracture is reliable (*p* = 0.029). Stable fractures present the lowest anteversion difference values, while reverse oblique fractures present the highest difference. *Conclusions:* Our study reveals that the percentage of rotational malalignment after the intramedullary nailing of intertrochanteric fractures remains high (31.1%), despite the proper use of radiographic imaging during the intraoperative fracture reduction. In most cases (91.3%), this malalignment appears to be a matter of internal overcorrection. A clear correlation between hip’s rotational deformity and patients′ functional outcome has yet to be proven, and constitutes our objective in the near future.

## 1. Introduction

Intertrochanteric femoral fractures are one of the most common causes of attendance at the orthopedic emergency department [1] and present a significant factor of morbidity and mortality—7.7% in the first 30 days and 26% within 1 year—for the affected patients [2]. The treatment of these fractures is usually operational, with closed reduction and internal fixation on a traction table [3]. Fracture reduction is particularly important for both the fracture’s union and patient’s postoperative functional outcome, as a poor fracture reduction can lead to a malposition, namely shortening, angulation, or a rotational deformity of the affected limb. The long-term clinical consequences of rotational malalignment after intramedullary nail fixation of intertrochanteric fractures are poorly known, as only a few studies targeting this issue have been conducted, none of which have been focused on the postoperative functional outcome. Such studies about rotational malalignment’s implications exist regarding the intramedullary nailing of diaphyseal and distal femoral fractures, in which degenerative arthritis as well as persistent pain and movement restriction of the hip and knee joints have been noted [4]. Although the quality of reduction in the frontal and sagittal planes can be easily assessed intraoperatively, the same is not true for the transverse plane, with rotational deformity being a frequent postoperative finding. Rotational malalignment is defined as the difference in anteversion between the operated and healthy hip. Femoral anteversion is defined as the projected angle between two lines, perpendicular to the femoral diaphysis in the transverse plane. One line passes through the axis of the femoral neck, while the other through the axis of the posterior femoral condyles. Rotational deformity alters the loads exerted on the femur and is associated with an increased incidence of functional complications. The postoperative evaluation of torsional deformity can be accomplished through clinical examination, radiography, ultrasound, and computed tomography, with the latter being the most accurate method [5]. The purpose of this study is to evaluate the rotational malalignment after intramedullary nailing for intertrochanteric fractures and to investigate a possible connection with specific patients’ or fractures’ characteristics.

## 2. Materials and Methods

A prospective study was conducted from September 2021 to July 2023, in a total of 74 patients presented to our emergency department, suffering from an intertrochanteric fracture. Patients with a history of a previous hip, femur, or knee operation, and those who were not able to undergo a CT scan, were excluded from this study.

Fracture type categorization was based on the AO/OTA classification system (stable, unstable, reverse oblique).

In the operating room, the patient was placed supine on a traction table, with the healthy lower extremity positioned in a lithotomy position and in 10–15° of abduction, under general or spinal anesthesia. The patient′s body was placed in a position of 10–15° abduction towards the healthy lower extremity. A closed reduction was performed on the fractured hip by applying traction along the patient’s longitudinal axis. Subsequently, the reduction quality was assessed radiographically using an image intensifier. In order to achieve a good reduction, the medial cortex continuity was assessed on the anteroposterior view, while the anterior cortical continuity was assessed on the lateral view (Figure 1). Any further adjustment to achieve an optimal reduction was done either by increasing/decreasing the applied traction force, or by applying further abduction/adduction or internal/external rotation to the affected limb. In the case that these adjustments were proven to be insufficient, fracture reduction was carried out via a mini-open approach with the help of a surgical tool (Hohmann retractor, Cobb elevator, reduction clump, bone hook).

All patients were treated with intramedullary nail fixation, by the same surgical team, using the same implant (Trigen Intertan Intertrochanteric Antegrade Nail, Smith & Nephew, Watford, England, UK).

Postoperatively, all of the patients underwent a CT scan of pelvis-hips and knees, if they were willing to and when they felt comfortable. On the CT scan, the anteversion of both the operated (angle 1) and the healthy (angle 2) hip was measured for every patient. The preferred method used for these measurements was the one suggested by Jeanmart et al. in 1983 [6] (Figure 2, Figure 3 and Figure 4). The difference (D angle) between the anteversion of the operated hip and the healthy hip was calculated. A D angle with a positive value indicated internal over-rotation of the distal fragment during reduction, while a D angle with a negative value indicated external over-rotation, respectively. The absolute value of the D angle represented the difference in hip anteversion.

Measurements were performed by one surgeon twice, and then once more by another surgeon. The mean intraobserver agreement was 1.4–1.9°, while the mean interobserver agreement was 1.5–2.1°.

Based on their calculated hip anteversion differences, the patients were divided into 3 groups. Patients with a D angle ≤ 5° were considered in the range of a normal difference in anteversion and were placed in group A. Group B included patients with an anteversion difference > 5° or <10°, which was considered as an acceptable rotational alignment. Finally, a D angle ≥ 15° represented a rotational deformity and each patient in this range was placed in group C.

Statistical analysis was done by using Student’s *t*-test for the continuous variables and the Pearson’s Chi-square test for the categorical variables. SPSS (version 27.0) was used for the statistical calculations. Statistical significance was determined at *p*-value < 0.05.

## 3. Results

From the 74 patients included in our study, 47 patients (63.5%) were females and 27 (26.5%) were males. The mean patients’ age was 80.6 years, with a range from 47 to 99 years. In 40 cases (54.1%), the injured side was the left one, while in the rest 34 cases (45.9%), the injured side was the right one. In most of the cases, namely 68 patients (91.9%), the cause of the fracture was a fall from standing height, while the remaining 6 patients (8.1%) suffered a high-energy injury (Table 1).

As for the fracture type, 28 cases (37.8%) were stable fractures, 35 cases were unstable fractures (47.3%), and 11 cases were reverse oblique fractures (14.9%). The mean anteversion on the operated hip was 23.2° with a standard deviation of 13.3° (ranging from 1.2° to 62.4°). On the healthy hip, the mean anteversion was 13.3° with a standard deviation of 7.2° (ranging from 2.2° to 36.7°).

A total of 42 patients (56.7%) presented with a normal anteversion difference lower than 5° (group A), 9 patients (12.2%) had an acceptable anteversion difference (group B) ranging higher than 5° but lower than 15°, and the remaining 23 patients (31.1%) showed a rotational deformity equal to or higher than 15° (Table 2).

A positive D angle value indicates an internal over-rotation of the distal fragment during fracture reduction, while a negative D angle indicates an external over-rotation, respectively. In a total of 74 patients, 59 (79.7%) presented with a positive D angle, with a mean value of 43° and a standard deviation of 33.3°, while 15 patients (20.3%) presented with a negative D angle, with a mean value of −21.3° and a standard deviation of 18.6° (Table 3).

The analysis of the correlation (Table 4) between the hip anteversion difference (absolute D angle value) and the rotational overcorrection (positive/negative D angle value) during fracture reduction shows that there is a statistically significant relationship between the two examined variables, as the statistical test values do exceed the thresholds of significance (*p* < 0.032).

The analysis of the correlation (Table 5) between the hip anteversion difference (absolute D angle value) and the fracture type shows that there is a strong relationship between these two variables. Stable fractures typically have smaller differences compared to unstable and reverse oblique fractures.

The analysis of the correlation (Table 6) between the hip anteversion difference (absolute D angle) and the need for reduction maneuvers indicates a statistically significant association or relationship between the variables (*p* < 0.036).

## 4. Discussion

Rotational deformity during the intraoperative reduction of an intertrochanteric fracture is one the most difficult complications to detect through an ordinary radiographic evaluation. This is because this type of evaluation only allows us to evaluate the quality of reduction in the frontal and sagittal planes, but is not sufficient to evaluate the rotational alignment of the fracture reduction in the transverse plane. Therefore, most of the existing studies focus more on valgus or varus deformities on the frontal plane and procurvatum or recurvatum deformities on the sagittal plane [7,8,9]. Regarding rotational deformities of the femur, these have been described in many studies, but usually for the intramedullary nailing of diaphyseal fractures [10,11,12]. At the same time, there are many studies that have been carried out which propose various ways to avoid rotational deformation in diaphyseal and distal femur fractures [13,14,15,16,17], but the ones regarding intertrochanteric fractures are very few and not well documented. The easiest and fastest way to diagnose a hip rotation deformity is usually a clinical examination, during which an internal or external rotation deformity of the operated limb against the healthy one can be detected. Nevertheless, clinical examination is, at the same time, the least valid way to establish the diagnosis of a torsional deformity, since it lacks both specificity and sensitivity [18,19]. Alternative methods to aid in the diagnosis of a hip rotational deformity are radiographic and ultrasound examination, which were the main means of diagnosing rotational deformities of the femur in the past; however, they are no longer used with the same frequency, since the measurements taken through them are complex and present with a low accuracy [18,20]. Nowadays, the most accurate and most frequently used way to diagnose torsional deformity of the femur is the use of the CT scan. Regarding the measurement criteria, many techniques have been described by various groups, but it has not been clearly demonstrated that any one of them is superior to the others, since they all show similar values of standard deviation and intraobserver–interobserver variability [21]. In the present study, we decided to use the technique described by Jeanmart et al. in 1983 [6], aiming to present comparable results with the few studies existing on this specific issue.

Our patients were divided into three groups based on the absolute value of the D angle, as derived from the difference in the anteversion measurements between the operated and healthy hip. A D angle ≤ 5° (group A) was considered a physiological difference in the anteversion, since the mean value of hip anteversion difference between the two hips in the general population is estimated to range from 2.9° to 4.1° in some studies [22] or from 5.1° to 8.8° in others [23]. On the contrary, according to the study by Jaarsma et al. [18], and all the other studies related to ours [24,25,26], a D angle ≥ 15° (group C) is considered as a rotational deformity. This definition is also in agreement with the results of other studies [23], according to which the maximum value of the difference in the anteversion between the two hips in the general population is estimated to be from 12° to 13°. Finally, a value of the D angle of less than 5° or higher than 15° (group B) was considered an acceptable rotational alignment. Table 7 shows the anteversion values (D angle) for the healthy and operated-on hips in all relative studies, while Table 8 shows the difference in the anteversion difference (absolute D angle), respectively.

According to the definition that we gave, the D angle expresses the difference between the operated-on and the healthy hip. Therefore, a D angle with a positive sign indicates the presence of excessive internal rotation in the distal segment during reduction, while a D angle with a negative sign indicates the presence of excessive external rotation, respectively. The absolute value of the D angle represents the difference in anteversion between the operated-on versus the healthy femur (Table 8). Table 9 presents the number and percentage of cases that presented rotational deformity (group C with D ≥ 15°), but also those for cases that presented excessive internal or external rotation, in the population of the present and other studies. In the present study, rotational deformity was observed in 31.1% of cases. The incidence rate is lower than the 40% of Ramanoudjame et al. [26], but slightly larger than the 25.7% of Kim et al. [25] and the 24.3% by Annappa et al. [24]. The mean rotational deformity was calculated to be 22.5° with a standard deviation of 6.1° and a range from 15° to 48.5°. The corresponding values for Ramanoudjame et al. [26] were 26.6° with a standard deviation of 10.3° and a range from 15.5° to 45°. Kim et al. [25] did not provide us with the corresponding data, while Annappa et al. [24] only provide us with the overall values of the anteversion difference (absolute angle D value) for the study population and not specific values for the group with the rotational deformity. Their overall sample mean was calculated to be 9.72° with a standard deviation of 5.78°.

As we can see from the abovementioned table (Table 9), rotational deformity after the intramedullary fixation of an intertrochanteric fracture is a relatively common complication, which results primarily from excessive internal over-rotation (leading to an increase in anteversion) in the distal fragment. This is in contrast to rotational deformity after the intramedullary fixation of a diaphyseal femoral fracture, which usually results from an excessive external over-rotation (reduction in anteversion) [27]. One possible explanation for this issue has been given by Jaarsma et al. [18], who attribute this condition to the anatomical location of the fracture. Specifically, in intertrochanteric fractures, the proximal fragment is subjected to flexion and external rotation due to the forces exerted on it by the iliopsoas and to adduction due to the corresponding forces from the gluteal and external rotator muscles [28]. Therefore, in order to achieve an optimal result during the reduction of an intertrochanteric fracture, we are required to counterbalance these forces by applying traction and internal rotation. Thus, according to the authors, rotational deformity in the form of internal rotation is caused by an overcorrection during fracture reduction. This theory is reinforced by the corresponding correlation in diaphyseal fractures. In these cases, the external rotational forces exerted by the hip muscles are reduced, in contrast to the medial rotational forces exerted by the thigh muscles, which are increased [28]. Compensation for these forces in a femoral diaphyseal fracture occurs by applying external rotation during reduction, so the usual external rotational deformity during the intramedullary nailing of these fractures can be attributed to an overcorrection analogous to that of the intertrochanteric fractures. In the present study, in most cases, it was deemed necessary to place the foot on the fractured limb in a position of 5–10° internal rotation after the closed reduction to achieve the optimal radiographic result. Placing the foot in internal rotation when positioning the patient on the traction table so that the patella appears centered is recommended by previous studies [29]. Nevertheless, this tendency could explain our results, since, in a total of 74 patients, 59 (79.7%) presented a greater value of anteversion (excessive internal rotation) in the operated-on hip compared to the healthy one. This percentage is even higher in cases in which rotational deformity was observed postoperatively, since, out of the 23 patients in group C, 21 (91.3%) presented with excessive internal rotation. In fact, similar results are presented in the research of Ramanoudjame et al. [26] (87.5%), who followed the same approach. In contrast, a slightly lower incidence rate of excessive internal rotation in the cases of rotational deformity is shown in the studies of Kim et al. [25] (67.8%) and Annappa et al. [24] (64.7%), who placed the extremity in a neutral position during fracture reduction. This leads us to the conclusion that positioning the foot in internal rotation should probably be avoided during the reduction of an intertrochanteric fracture, even if it gives us the best radiographic assessment in the frontal and sagittal planes intraoperatively, since the rotational alignment of the reduction of the fracture in the transverse plane is often not the desirable. A solution to this issue could potentially be the radiological protocol proposed by Tornetta et al. [30]. According to this protocol, at first, we perform a radiographic evaluation of the anteversion on the healthy hip and then the fracture reduction is performed based on that anteversion angle. Of course, according to the authors, this procedure can increase the surgical time by at least 15 min, while the possibility of its correct execution by non-trained radiologists and surgeons is also uncertain. Several alternative methods have been described, but most are characterized by a low sensitivity [31].

Regarding the characteristics of our sample, the measurements show that factors such as the age and gender of the patient, the side of the affected hip, and the mechanism of injury do not show any correlation with the complication of the rotational deformity. On the contrary, the type of fracture seems to play a catalytic role in the outcome, since the statistical significance for the correlation of the mean D angle value for the three types of fractures is reliable (*p* = 0.029). Specifically, patients with stable fractures present a mean value of hip anteversion difference of 9.1°, those with unstable fractures a mean value of 12.9°, while, for those with reverse oblique fractures, the mean value is calculated at 18.6°. Of the existing studies, only that of Annappa et al. [24] analyzed the correlation between the rotational malalignment and fracture type, ending up with results similar to ours. This is something that can be explained by the nature of the fracture. In stable fractures, reduction is usually an easier process, since the integrity of the posterior cortex acts as a stable ‘hinge’, thus aiding in the reduction of the displaced anterior cortex. However, in unstable and reverse oblique fractures, reduction is almost always a more demanding procedure. Fragmentation of the posterior cortex removes the stabilizing factor of the ‘hinge’, so the rotational manipulations required to reduce the anterior cortex may have exactly the opposite effect in the posterior cortex. In such cases, further maneuvers, via a mini open approach and by using an appropriate surgical tool, are usually deemed necessary to achieve an optimal reduction. This is also confirmed by our results, which show strong statistical evidence for the relationship between the type of fracture and the necessity of reduction maneuvers via a mini open approach. Out of the 11 total cases in which further reduction maneuvers were required, 6 (54.5%) belonged to reverse oblique fractures, another 3 cases (27.3%) belonged to unstable fractures, and only 2 cases (18.2%) were cases of stable fractures. However, despite the fact that these maneuvers help us in achieving good radiographic reduction intraoperatively, our measurements prove that they are not always sufficient to correct the rotational malalignment. In a total of 11 patients in whom further reduction maneuvers were applied intraoperatively, 7 (63.6%) eventually developed a rotational deformity, with our analysis confirming a strong statistical association.

We know that the very important part of the clinical follow-up is missing from this study, but this is an objective for us in the near future, as right now our patients’ follow-up is underway. To our knowledge, the only study reaching this topic to some extent is the study of Kim et al. [25]. We aim to monitor the patients’ functional outcomes, the possible complications, and to explore a possible connection between patients’ mortality rate and the rotational deformity after the intramedullary nailing of intertrochanteric fractures.

## 5. Conclusions

Despite a thorough intraoperative radiographic evaluation of the reduction quality, our study reveals that this alone may not be sufficient to prevent the high percentage (31.1%) of rotational malalignment after the intramedullary nailing of intertrochanteric fractures. In most cases (91.3%), this rotational malalignment appears to be a matter of internal overcorrection during fracture reduction. Although a clear correlation between hip rotational deformity and patients’ functional level has yet to be proven, new assessment methods to prevent such a complication must be established. The use of intraoperative 3D imaging or computer-assisted navigation might be worth examination as areas for future development. Additionally, our future focus turns towards the possible effect of such a complication on the patients’ functional outcome.

## Figures and Tables

**Figure 1 medicina-60-01535-f001:**
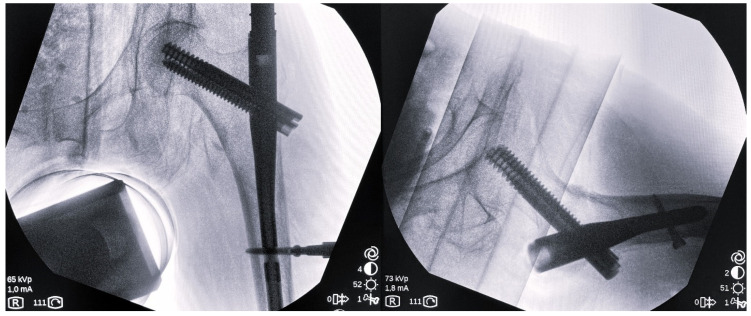
Anteroposterior (**left**) and lateral (**right**) view of an intraoperative radiographic evaluation after the fixation of the intramedullary nail.

**Figure 2 medicina-60-01535-f002:**
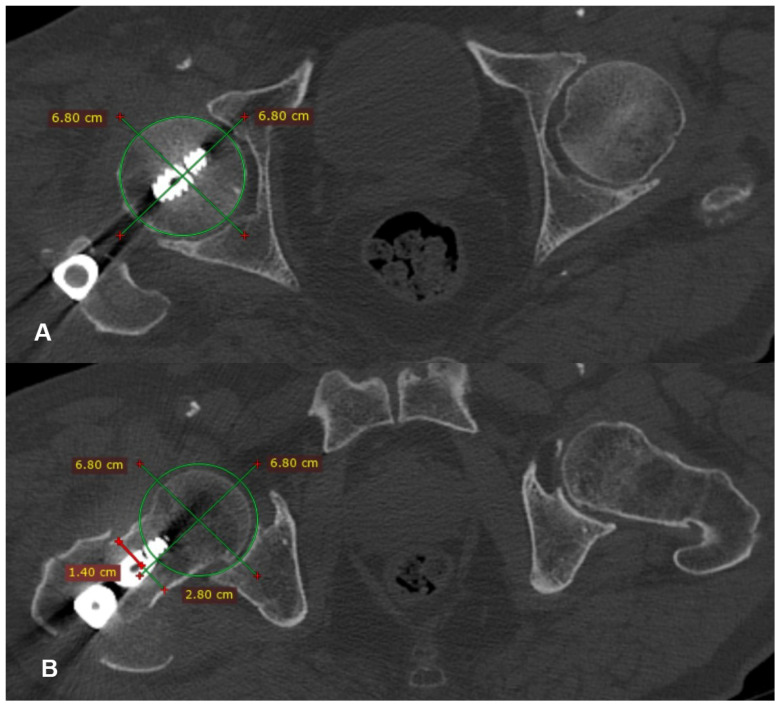
Determining the center of the femoral head in the transverse section where it presents the maximum diameter (**A**). Determining the center of the femoral neck in the transverse section where it presents its narrowest width (**B**).

**Figure 3 medicina-60-01535-f003:**
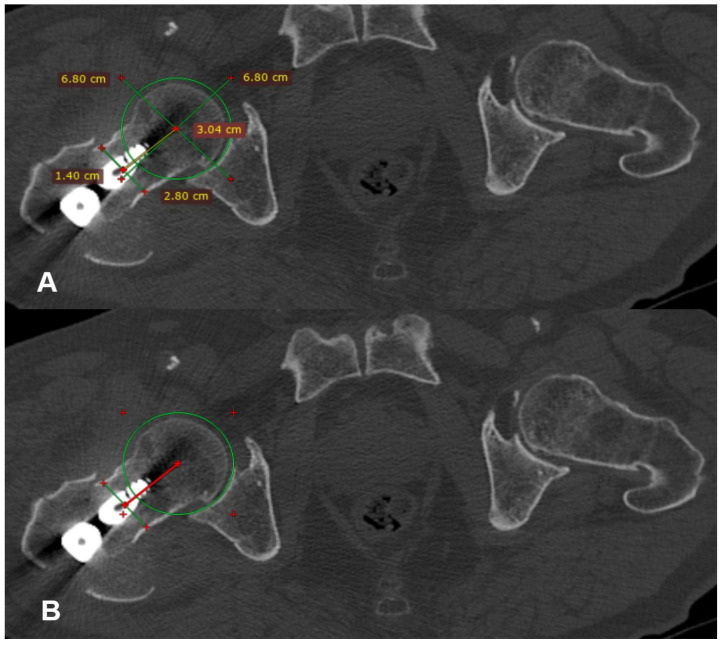
A line is drawn which connects the center of the head with the center of the neck (yellow line (**A**), red line (**B**)).

**Figure 4 medicina-60-01535-f004:**
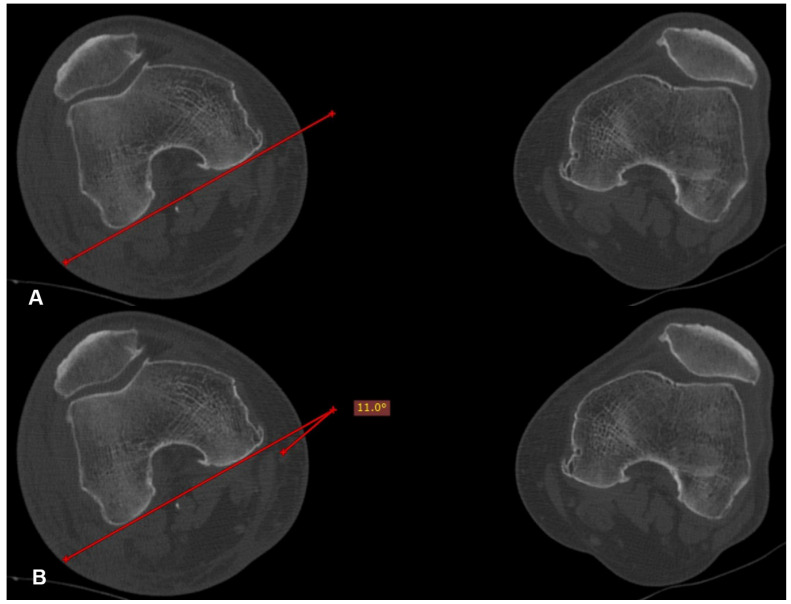
Tangent line to the posterior femoral condyles in the transverse section where they present the maximum anteroposterior diameter (**A**). Projective angle formed between the tangent line at the posterior femoral condyles and the line connecting the center of the head with the center of the neck. This angle expresses the femoral anteversion (**B**).

**Table 1 medicina-60-01535-t001:** Incidence and relative incidence of sample’s characteristics and study’s results.

		*N*	%
Gender	Male	27	36.5%
	Female	47	63.5%
Side	Right	34	45.9%
	Left	40	54.1%
AO/OTA Fracture Classification	Stable	28	37.8%
	Unstable	35	47.3%
	Reverse Oblique	11	14.9%
Reduction Maneuvers	Yes	11	14.9%
	No	63	85.1%
Injury Mechanism	Low-Energy	6	8.1%
	High-Energy	68	91.9%

**Table 2 medicina-60-01535-t002:** Analysis of anteversion difference between operated and healthy hip for each patient group.

Group	D Angle (Absolute Value)	*N*	%	Minimum Value	Mean Value	Standard Deviation	Range	Maximum Value
A	≤5°	42	56.7%	0.9°	2.6°	1.4°	5°	4.9°
B	>5° to <15°	9	12.2%	5.2°	9.6°	2.2°	9.6°	14.8°
C	≥15°	23	31.1%	15°	22.5°	6.1°	33.5°	48.5°
Total		74	100.0%	0.9°	12.3°	10.3°	47.6°	48.5°

**Table 3 medicina-60-01535-t003:** Analysis of the internal/external rotational overcorrection during fracture’s reduction.

		*N*	%	Mean	Standard Deviation
D Angle	Negative	15	20.3%	−21.3	18.6
	Positive	59	79.7%	43.0	33.3

**Table 4 medicina-60-01535-t004:** Frequency and relative frequency of cross table between hip anteversion difference (absolute D angle) and internal/external rotational overcorrection (D angle).

			Group			Total
			A	B	C	
D angle	Positive	*N*	32	6	21	59
		%	76.2%	66.7%	91.3%	79.7%
	Negative	*N*	10	3	2	15
		%	23.8%	33.3%	8.7%	20.3%
Total		*N*	42	9	23	74
		%	100.0%	100.0%	100.0%	100.0%

**Table 5 medicina-60-01535-t005:** Correlation between hip anteversion difference (absolute D angle) and fracture type.

AO/OTA Classification	*N*	Mean Value	Standard Deviation	Standard Error	Minimum Value	Maximum Value	F	Sig.
Stable Fracture	28	9.1°	8.9°	1.7°	0.9°	37.1°	3.737	0.029
Unstable Fracture	35	12.9°	10.2°	1.7°	1.3°	48.5°		
Reverse Oblique Fracture	11	18.6°	11.3°	3.4°	1.7°	33.0°		
Total	74	12.3°	10.3°	1.2°	0.9°	48.5°		

**Table 6 medicina-60-01535-t006:** Frequency and relative frequency of cross table between hip anteversion difference (absolute D angle) and use of reduction maneuvers.

			Group			Total
			A	B	C	
**Reduction Maneuvers**	**Yes**	*N*	4	0	7	11
		%	9.5%	0.0%	30.4%	14.9%
	**No**	*N*	38	9	16	63
		%	90.5%	100.0%	69.6%	85.1%
**Total**		*N*	42	9	23	74
		%	100.0%	100.0%	100.0%	100.0%

**Table 7 medicina-60-01535-t007:** Comparison of the anteversion of the operated and healthy hip between studies.

Examined Hip	Study	Minimum Value	Mean Value	Standard Deviation	Maximum Value
Operated-on	Ramanoudjame et al. [26]	−33°	23°	16.8°	47°
	Kim et al. [25]	-	-	-	-
	Annappa et al. [24]	-	15.7°	8°	-
	Present study	1.2°	23.2°	13.3°	62.4°
Healthy	Ramanoudjame et al. [26]	5°	14.2°	5.6°	25.1°
	Kim et al. [25]	−6.4°	11.7°	-	37°
	Annappa et al. [24]	-	13.2°	9.4°	-
	Present study	2.2°	13.3°	7.2°	36.7°

**Table 8 medicina-60-01535-t008:** Comparison of the difference in operated-on versus healthy hip anteversion (absolute angle D value) in existing studies.

Study	Minimum Value	Mean Value	Standard Deviation	Maximum Value
Ramanoudjame et al. [26]	1.4°	15.3°	11.7°	45°
Kim et al. [25]	-	20.7°	-	-
Annappa et al. [24]	-	9.7°	5.8°	-
Present study	0.9°	12.3°	10.3°	48.5°

**Table 9 medicina-60-01535-t009:** Comparison of postoperative rotational malalignment with other studies.

Study	Ν	Rotational Malalignment	Internal Overcorrection	External Overcorrection
Ramanoudjame et al. [25]	40	16 (40%)	14 (35%)	2 (5%)
Kim et al. [24]	109	28 (25.7%)	19 (17.4%)	9 (8.3%)
Annappa et al. [23]	70	17 (24.3%)	11 (15.7%)	6 (8.6%)
Present study	74	23 (31.1%)	21 (28.4%)	2 (2.7%)

## Data Availability

All raw data are available to access should they be requested.

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
