# Peer review of "Incidence of Rotational Malalignment after Intertrochanteric Fracture Intramedullary Nailing: A CT-Based Prospective Study"

_medicina, 2024, doi:10.3390/medicina60091535_

Round 1
Reviewer 1 Report
Comments and Suggestions for Authors
The introduction could benefit from more in-depth contextualization of the clinical implications of rotational malalignment.
The section describing the radiological protocol and the calculation of the anteversion angles would benefit from further clarity regarding how variability between observers was managed. Although there is mention of two different surgeons performing measurements, it would be useful to explicitly state whether inter-observer variability was calculated and how discrepancies were handled.
The discussion around rotational deformities could be enhanced by integrating a more thorough exploration of how this deformity correlates with patient outcomes.
While the study's conclusion outlines the need for new assessment methods, the authors could discuss potential future techniques that could address the issue. For example, the use of intraoperative 3D imaging or computer-assisted navigation might be worth mentioning as areas for future development.
minor revisions
Author Response
Dear reviewer 1,
First of all, thank you for taking the time to review our paper; we are glad you found it interesting.
Comment 1: The introduction could benefit from more in-depth contextualization of the clinical implications of rotational malalignment.
Response 1: Our introduction lacks more information on clinical implications as no studies have been performed focusing on this. In fact, our own post-operational functional outcome study on our study’s population is underway, and we intend to publish our results regarding this issue soon. Following your advice, we noted this and we added some clinical results from malalignment studies on intramedullary nailing of diaphyseal and distal femoral fractures.
Comment 2: The section describing the radiological protocol and the calculation of the anteversion angles would benefit from further clarity regarding how variability between observers was managed. Although there is mention of two different surgeons performing measurements, it would be useful to explicitly state whether inter-observer variability was calculated and how discrepancies were handled.
Response 2: Mean values for intra- and interobserver agreement were included in the materials and methods.
Comment 3: The discussion around rotational deformities could be enhanced by integrating a more thorough exploration of how this deformity correlates with patient outcomes.
Response 3:
About the discussion regarding patient outcomes applies the same as for the introduction part. Current literature has almost no evidence on this topic and we aim to publish our own result soon.
Comment 4: While the study's conclusion outlines the need for new assessment methods, the authors could discuss potential future techniques that could address the issue. For example, the use of intraoperative 3D imaging or computer-assisted navigation might be worth mentioning as areas for future development.
Response 4: It was really helpful of you to point out that we were missing potential future techniques that could address this issue, so we added your suggestions to our conclusion.
Reviewer 2 Report
Comments and Suggestions for Authors
The article is interesting and original, but some points need to be reviewed before publication:
- The introduction section too short, it could be improved. This is a vast field for attracting readers' attention.
- The references cited could be updated. Complemented;
- The material and methods section needs to be cited, especially the evaluations carried out.
- Human study ethics committee? It needs to be mentioned.
- The discussion could explore the literature further by checking the findings of the present study against the data in the literature.
What are the future perspectives?
- Throughout the manuscript, many sections need to be referenced. Above all, the references can be updated. Many should be replaced by recent ones.
Author Response
Dear reviewer 2,
First of all, thank you for taking the time to review our paper; we are glad that you found it interesting.
Comment 1: The introduction section too short, it could be improved. This is a vast field for attracting readers' attention.
Response 1: We updated our introduction section by adding a paragraph regarding the functional impact of rotational malalignment in intertrochanteric fracture intramedullary nailing.
Comment 2: The material and methods section needs to be cited, especially the evaluations carried out.
Response 2: The technique used on CT scans to evaluate our results have been cited. All the other stuff described in this section are the things that we did (patients position during the operation, use of surgical instruments, etc.) in order to conduct our study. Dividing our population in groups based on the anteversion value has been cited in the discussion section.
Comment 3: Human study ethics committee? It needs to be mentioned.
Response: Human study committee approval has been provided to the editor.
Comment 4: The discussion could explore the literature further by checking the findings of the present study against the data in the literature.
Response 4: In our discussion we compared our results with the ones of all the 3 studies that exist in literature regarding this topic. In fact, we also provided some tables to making these comparison more clear.
Comment 5: What are the future perspectives?
Response 5: Future techniques that can be used intraoperatively to avoid this issue have been added to the conclusion.
Comment 6: Throughout the manuscript, many sections need to be referenced. Above all, the references can be updated. Many should be replaced by recent ones.
Response: We really tried to update our references and replace them with recent ones, but most of them are unique about this topic or described innovative techniques at their publication time that are still used today.